# Spinal Cord Involvement in MS and Other Demyelinating Diseases

**DOI:** 10.3390/biomedicines8050130

**Published:** 2020-05-22

**Authors:** Mariano Marrodan, María I. Gaitán, Jorge Correale

**Affiliations:** Neurology Department, Fleni, C1428AQK Buenos Aires, Argentina; mmarrodan@fleni.org.ar (M.M.); migaitan@fleni.org.ar (M.I.G.)

**Keywords:** myelitis, spinal cord, multiple sclerosis, neuromyelitis optica, acute disseminated encephalomyelitis, myelin oligodendrocyte glycoprotein, glial fibrillary acidic protein

## Abstract

Diagnostic accuracy is poor in demyelinating myelopathies, and therefore a challenge for neurologists in daily practice, mainly because of the multiple underlying pathophysiologic mechanisms involved in each subtype. A systematic diagnostic approach combining data from the clinical setting and presentation with magnetic resonance imaging (MRI) lesion patterns, cerebrospinal fluid (CSF) findings, and autoantibody markers can help to better distinguish between subtypes. In this review, we describe spinal cord involvement, and summarize clinical findings, MRI and diagnostic characteristics, as well as treatment options and prognostic implications in different demyelinating disorders including: multiple sclerosis (MS), neuromyelitis optica spectrum disorder, acute disseminated encephalomyelitis, anti-myelin oligodendrocyte glycoprotein antibody-associated disease, and glial fibrillary acidic protein IgG-associated disease. Thorough understanding of individual case etiology is crucial, not only to provide valuable prognostic information on whether the disorder is likely to relapse, but also to make therapeutic decision-making easier and reduce treatment failures which may lead to new relapses and long-term disability. Identifying patients with monophasic disease who may only require acute management, symptomatic treatment, and subsequent rehabilitation, rather than immunosuppression, is also important.

## 1. Introduction

Diagnostic accuracy in myelopathies is poor and therefore a challenge for neurologists in daily practice, mainly due to the multiple underlying pathophysiologic mechanisms observed in this group of disorders. In an initial approach, temporal profile (time to symptom nadir) contributes to differentiate vascular or traumatic causes from those of metabolic, neoplastic, and infectious or inflammatory etiology. To further assist in the identification of patients with acute vascular myelopathies for whom specific treatment strategies may be indicated, patients whose symptoms reach maximal severity in <4 h from onset are currently presumed to have an ischemic pathology unless proven otherwise [1]. By contrast, inflammatory processes affecting the spinal cord produce symptoms in a subacute manner, typically over hours or days. However, despite extensive patient work-up, a significant number of myelopathy cases are ultimately considered idiopathic [2]. Unfortunately, the term inflammatory myelitis is still applied to a complex and heterogeneous subgroup of post-infectious, rheumatologic, granulomatous, paraneoplastic, and demyelinating diseases, commonly affecting the spinal cord in which substantial overlap in clinical and imaging findings subsists. Identifying relapsing forms of disease has prognostic implications and can guide preventive treatment. Failure to indicate appropriate treatments may lead to new relapses and long-term disability. In contrast, patients in whom monophasic disease is suspected may only require acute management, symptomatic treatment, and subsequent rehabilitation rather than immunosuppression. In the case of demyelinating disorders, although multiple sclerosis (MS) is the main cause of inflammatory myelitis, other important differential diagnoses need to be ruled out to select the best treatment strategy in individual patients [3,4]. Thorough understanding of individual case etiology is therefore crucial, not only for correct treatment, but also to determine patient outcome.

In this review, we describe the epidemiologic characteristics, pathophysiology, clinical and (magnetic resonance imaging) MRI findings, treatment options and prognostic implications in MS and other demyelinating disorders including: neuromyelitis optica spectrum disorder (NMOSD), acute disseminated encephalomyelitis (ADEM), anti-myelin oligodendrocyte glycoprotein (MOG)-antibodies (ab) associated disease, and glial fibrillary acidic protein (GFAP)-IgG associated disease, to provide guidance in the diagnosis of these conditions.

A Pubmed search was conducted for articles published between 2000 and 2020, that included the terms: “acute disseminated encephalomyelitis; “demyelinating diseases”; “glial fibrillary acidic protein”; “multiple sclerosis”; “myelin oligodendrocyte glycoprotein”; “myelitis”; “neuromyelitis optica”; and “spinal cord diseases”. Only those originally in English were considered. Earlier publications were identified from references cited in the articles reviewed.

## 2. Multiple Sclerosis

MS is a chronic inflammatory disease of the CNS leading to demyelination, neurodegeneration, and gliosis. It is by far the most common demyelinating disease, affecting over 2 million people worldwide [5]. Although its etiology remains elusive, environmental factors and susceptibility genes are now known to be involved in the pathogenesis [6]. Results from immunological, genetic, and histopathology studies of patients with MS support the concept that autoimmunity plays a major role in the disease [7]. In the majority of cases, the disease follows a relapsing remitting course (RRMS) from onset, which may later convert into a secondary progressive form (SPMS). Less often, patients show continued progression from disease debut (primary progressive MS, PPMS) [8].

Spinal cord abnormalities are common in MS and include a variety of pathological processes, such as demyelination, neuroaxonal loss and gliosis. Ultimately these result in motor weakness with accompanying difficulties in deambulation, spasticity, sensory disturbances, as well as bladder and bowel dysfunction [9]. Relapsing remitting MS can cause acute myelitis presenting with sensory loss, gait impairment, and incoordination, generally worsening over days to weeks, followed by stabilization or recovery [10]. During progressive phases of the disease however, especially in PPMS, slowly increasing or stuttering gait impairment due to demyelinating myelopathy is the most frequent presentation [11]. Once gait impairment has developed, cumulative disability increase will depend on patient age, clinical, and radiological disease activity and degree of spinal cord atrophy [12,13,14,15].

Histopathology findings in the spinal cord are characterized by significant decrease in axonal density in normal-appearing white matter (NAWM); perivascular T-cell infiltrates are rare, but robust, and diffuse inflammation is observed both in normal-appearing parenchyma and particularly in the meninges. Extent of diffuse axonal loss in NAWM correlates with both MHC class II^+^ microglia cell density in NAWM, and significant increase in T cell density in the meninges. Interestingly, close interaction has been observed between T cells and MHC class II^+^ macrophages in spinal cord meninges from MS patients, suggesting the meninges may form an immunological niche in which T lymphocytes become activated and proliferate in response to antigen presentation [16]. In support of this concept, similar findings have previously been described in experimental autoimmune encephalomyelitis [17], raising the possibility that activated meningeal T cells, through release of soluble factors such as IFN-γ, could instruct parenchymal macrophages/microglia to engage in neurotoxic activation programs [18].

Although spinal cord involvement has been difficult both to characterize and to quantify because current clinical and MRI parameters lack sensitivity and specificity [19], the spinal cord was one of four anatomical locations incorporated in a revision to McDonald diagnostic criteria for MS in 2017, to document spatial dissemination in patients presenting clinical isolated syndrome (CIS) suggestive of MS. Likewise, new or gadolinium-enhancing spinal cord lesions can be used to document chronological progression [20].

Poor correlation between spinal cord injury load and clinical disability may be due to several different factors. Spinal cord MRI is more challenging than brain imaging in patients with MS. The spine is extremely thin and commonly subjected to ghosting artifacts (due to breathing, swallowing, and/or pulsation of blood and cerebrospinal fluid (CSF)) [21]. The amount of bone and fat may also produce significant artifacts, greater than those observed in brain imaging. Conventional, sagittal proton density (PD) and T2-weighted scans, with spatial resolution of 3 × 1 × 1 mm, should be considered the reference standard to detect MS spinal cord lesions [22,23]. Short-tau inversion recovery (STIR) sequences seem to be more sensitive to lesion detection than T2-weighted sequences and may be used to substitute PD sequences [24]. Contrast-enhanced T1-weighted images are recommended if T2 lesions are detected.

Conventional spine MRI has low sensitivity and specificity in relation to the pathological changes observed in MS [25]. Use of sagittal sections alone may underestimate lesion numbers [25]. Axial imaging may detect more lesions than sagittal imaging [26], especially smaller ones in the spinal cord periphery [27] and 2D or 3D T2-weighted sequences should be included in MRI protocols [21]. Axial multiple-echo recombined gradient echo (MERGE) seems to provide greater sensitivity for cord lesion detection and may represent a good alternative [28]. Ultimately, combined use of sagittal and axial images can facilitate identification and location of spinal lesions (Figure 1A–F) [26].

Often more than one demyelinating plaque is present in spinal cord MRIs from patients with MS. The cervical spine (53–59%) is the most common location, followed by the thoracic region (20–47%) [10]. Lesions usually present as hyperintense on T2-weighted and isointense on T1-weighted sequences. Gadolinium enhancement is variable and depends mainly on acquisition timing, with acute lesions usually enhancing during 4–8 weeks [29,30]. Most MS lesions are small in size, wedge-shaped in axial and ovoid-shaped in sagittal views, and predominantly found in ascending sensory (i.e., posterior column), and descending motor (i.e., corticospinal) spinal cord tracts, because of the high myelin concentration within these fascicules [31]. Rarely, they may extend to involve central grey matter, occupying over half the cross-sectional area of the cord. Small focal lesions may coalesce to form more extensive ones, involving three or more segments, particularly in cases of progressive MS. High-resolution axial MRI demonstrates these images actually result from the confluence of multiple discrete lesions [25,32].

Spinal cord lesions, when present, are particularly helpful to discriminate MS from its radiological mimics, which include conditions such as migraine and cerebrovascular disorders. They can also present together with multifocal T2 lesions in brain white matter [33].

In addition to their diagnostic value, spine lesions contribute prognostic information in MS. Asymptomatic lesions are present in approximately 35% of patients with radiological isolated syndrome [34], in one-third of patients with CIS [35], and 83% of patients with early RRMS [36]. Interestingly, the number of asymptomatic lesions found in patients with CIS has been linked to risk of a second clinical event at 2 and 5 years [37,38], making spine MRI advisable in CIS patient workup. However, detection of asymptomatic spinal cord lesions during follow-up of RRMS patients was less common than detection of asymptomatic lesions in the brain, suggesting spinal cord MRI may be less useful than brain MRI for monitoring patients with RRMS [39]. Some authors have observed that greater number of spinal cord lesions at MS time of diagnosis and lesional topography at time of relapse were associated with increased relapse rates and higher risk of developing secondary progressive MS [10,11,39].

Spinal cord atrophy (Figure 1G) present in early stages of the disease may correlate with degree of disability and predict long term outcome [38,40]. Measuring changes in cross-sectional area at the cervical level yields the most reproducible results and shows closest correlation to clinical findings [41,42]. Grey matter atrophy on the other hand correlates more strongly with degree of physical disability than other MRI parameters of brain and cord atrophy [43,44,45]. Notably, a significant association between reduced cervical cord sectional diameter and disability progression has been demonstrated in different studies, independent of brain atrophy [46,47,48]. Cord atrophy has also been associated with reduction in retinal nerve layer thickness [48], suggesting it is probably part of a global pathological process and not just determined by local damage. 

Rate of atrophy is more accelerated in the spinal cord than in the brain (1.5–2.2% per year vs. 0.5–1% per year) [49,50], and in patients with SPMS than in patients with CIS or RRMS. In RRMS, cord atrophy presents primarily in the posterior spinal cord, while in SPMS, atrophy is generalized [49]. Interestingly, regional atrophy does not seem to be influenced by focal lesion presence [51,52,53]. A recent study reported that a 1% increase in the annual rate of spinal cord volume loss was associated with a 28% risk of disability progression in the subsequent year [50]. Unfortunately, widespread use of this parameter has so far been limited by poor reproducibility and lack of sensitivity to small changes in the cord cross-sectional area. Since the rate of spinal atrophy over time appears to be associated with disability progression, atrophy has been considered a secondary outcome measure in phase 3 clinical trials of progressive MS [50,51]. When it was later analyzed more thoroughly, results were inconclusive [54,55,56]. This may have been due to lack of treatment efficacy, inadequate patient selection, poor reproducibility of cord atrophy quantification, or low sensitivity of MRI techniques used to detect small changes in cord cross-sectional area [30]. Eventually, spinal cord atrophy could also be considered a primary outcome in phase 2 clinical trials of progressive MS. However, this will require adequate patient selection and more precise MR imaging techniques for exact assessment.

It should be noted that neuronal loss in MS is not limited to white matter only, post mortem studies have also shown extensive neuronal loss in gray matter of the spinal cord as well, generating considerable interest in detection of gray matter abnormalities in MS [57]. The use of a combination of axial fast-field echo (FFE) and phase-sensitive inversion recovery (PSIR) sequences has been proposed to identify gray matter abnormalities in the upper cervical spinal cord [58]. However, further studies are still necessary to verify the sensitivity of this technique. Although both double inversion recovery (DIR) and PSIR can help distinguish focal gray matter lesions from normal-appearing tissue on sagittal views [59,60], these are often confounded by artifacts [61,62]. Use of 3T and higher field-strength (4.7 T) scanners as well as dedicated imaging sequences have increased MRI sensitivity for MS-associated gray matter lesion detection in the spinal cord [63,64]. Nevertheless, greater knowledge of spinal cord lesion pathogenesis, as well as its relationship to disability progression, still need to be established to better define the role of spinal cord assessment in MS diagnosis and follow-up [30].

More sensitive and better standardized methods are needed to assess clinical manifestations related to spinal cord atrophy over time, as well as monitor disease course and response to therapy. Promising MRI techniques to study the spinal cord include myelin water imaging, magnetization transfer imaging, diffusion tensor imaging, and magnetic resonance spectroscopy. At present however, use of these modalities is mostly restricted to research. Automated image-acquisition techniques, increased precision, and reduced quantification variability over time still need to be developed, and application in the clinical setting will likely be limited to select sites with experience using advanced imaging techniques. 

Several studies have demonstrated that residual deficits persist after MS relapses affecting the spinal cord, contributing to stepwise progression of disability. For this reason, prompt and adequate treatment of relapses is key, although optimal regimens have to be better defined [65,66]. Unfortunately, despite significant advances in disease-modifying treatment, management of acute MS relapses with intravenous or oral corticosteroids has remained largely unchanged for the past 20 years [67]. 

Since the first prospective trial demonstrated superiority of high-dose intravenous methylprednisolone use (IVMPS; up to 1000 mg daily) over placebo, acute MS relapses are initially treated with IVMPS during three to five days [68]. Although faster recovery of relapses has been documented, clinical improvement is insufficient in approximately 25% of patients after the first course of IVMPS [69]. Aside from increasing steroid treatment dose and prolonging treatment (up to 2000 mg daily for five additional days), use of plasma exchange (PLEX) has also been considered an alternative option [70]. One recent study in a group of patients receiving PLEX within 6 weeks of a relapse showed not only significantly better response rates than those of patients receiving extended IVMPS treatment, but also lower risk deterioration 3 months after discharge [71,72]. For long-term treatment of MS, the last 2 decades have seen the development of numerous drugs aimed at correcting the different pathogenic mechanisms proposed in multiple sclerosis, most of which have been compounds targeting immune system dysfunction. Several clinical trials are currently ongoing, some using neuroprotective therapies to halt progression, others aimed at reversing neurological disability, at least in part, by repairing damaged brain and spinal cord tissue. Discussion of particular disease-modifying therapies for MS is beyond the scope of this manuscript, however, several comprehensive reviews on the subject have recently been published [73,74,75,76].

## 3. Acute Disseminated Encephalomyelitis

Acute disseminated encephalomyelitis (ADEM) is an autoimmune demyelinating disorder of the CNS, commonly affecting brain and spinal cord white matter, although deep grey matter nuclei (e.g., thalamus and basal ganglia) may also be involved [77,78]. ADEM is more common in children (mean age 5 to 8 years), but can occur at any age [79] with an estimated annual incidence of 0.23 to 0.40/100,000 children [80,81,82]. Although no clear gender predominance has been identified, slight male preponderance has been described in some pediatric ADEM cohorts [79]. Most pediatric ADEM cases appear to be preceded by symptoms of viral or bacterial infection, usually of the upper-respiratory tract. Vaccination has also been reported to precede ADEM, although at much lower rates [83]. Some cases have been linked to specific vaccines produced in neural tissue cultures (rabies and Japanese B encephalitis). However, a marked drop in post vaccination ADEM has occurred since CNS tissue culture-derived production was replaced by recombinant protein-based vaccines. Nevertheless in up to 26% of patients, no triggering event can be observed [84].

Histopathology findings in ADEM show perivenular inflammatory infiltrates consisting of T cells and macrophages, associated with perivenular demyelination and relative preservation of axons in most cases. In hemorrhagic variants, demyelination is often more widespread through the CNS, with important neutrophilic infiltrates [79].

The pathogenesis of ADEM is still unclear. Two main hypotheses have been proposed. One, the molecular mimicry hypothesis, suggests partial structural or amino-acid sequence homology may exist between certain pathogens or vaccines and host CNS myelin antigens, which in turn may activate myelin-reactive T cells, thereby eliciting a CNS-specific autoimmune response [85]. The second hypothesis proposes CNS infection may directly prompt a secondary inflammatory cascade, leading to blood-brain barrier rupture, exposure of CNS-antigens, and breakdown of tolerance resulting in an autoimmune attack driven mainly by T cells [86]. 

Criteria for ADEM diagnosis, established in 2013 by the International Multiple Sclerosis Study Group (IPMSSG), require the following to be present: (1) an initial polyfocal clinical CNS event of presumed inflammatory demyelinating cause; (2) encephalopathy (alteration in consciousness or behavior unexplained by fever, systemic illness, or post ictal symptoms); (3) brain MRI abnormalities consistent with demyelination during the acute phase (first 3 months); (4) no new clinical or MRI findings 3 months or more after onset [87].

Depending on the series, spinal cord involvement has been described in 20% to 54% of ADEM patients, predominantly affecting the thoracic region [88]. Coincident brain and spinal cord lesions are more common; isolated spinal cord ADEM is exceptional [89] and typically extends over multiple segments, cause cord swelling, and showing variable enhancement in the acute phase. In most ADEM patients, partial or complete resolution of MRI abnormalities occurs within a few months of treatment [84,90]. Interestingly, ADEM patients with anti-MOG antibodies show large, more widespread brain lesions with ill-defined borders and longitudinally extensive spinal cord lesions on MRI [91]. Lesions involving more than two segments are more frequent in adults than in children (50% vs. 27%, respectively) [92]. 

No specific studies on CSF have been conducted in ADEM. Pleocytosis is typically mild, with a high percentage of lymphocytes and monocytes [92,93] and increased protein levels (up to 1.1 g/L) in 23% to 62% of pediatric patients [94,95,96]. OCBs are only present in 0% to 29% of cases [79]. However, they are usually transient as opposed to those observed in MS. 

Although ADEM usually has a monophasic course, multiphasic forms have been reported in 10–31% of patients [84,97], making differential diagnosis with MS more difficult in these cases. Multiphasic forms are defined as new encephalopathic events consistent with ADEM, separated by a 3-month interval from the initial illness but not followed by any further event [98]. Relapsing disease following ADEM occurring beyond a second encephalopathic event is no longer consistent with multiphasic ADEM, but rather indicates a chronic disorder such as MS, NMOSD, or ADEM-optic neuritis [98,99], and should prompt testing for anti-MOG ab. It is worth highlighting that progression from ADEM to MS is relatively low, estimated at 0% to 17% in studies with follow-up periods lasting several years [88].

Clinical presentation and outcome of ADEM in adults differs from that of children. Disease course is worse in adults, with more than one-third of patients requiring admission to an ICU, and duration of hospitalization can be twice as long. Outcome is also less favorable, complete motor recovery is observed in only 15% of adults compared to 58% of children and more adult patients die, although no difference in the occurrence of relapses or conversion to MS has been reported [92,100]. Poorer outcomes in adults cannot be explained by differences in clinical presentation (preceding factors, symptoms, blood and CSF parameters or radiological features are all similar). Perhaps reduced plasticity in ageing CNS tissue is the cause, rather than a difference in pathophysiology from onset [92].

No randomized-controlled studies have been conducted on ADEM treatment. Despite the lack of conclusive evidence, a widely accepted regimen in use today is administration of IV methylprednisolone (30 mg/kg/day in children or 1000 mg/day in adults) for 5 days, followed by oral taper with dexamethasone at a starting dose of 1–2 mg/kg/day, for 4–6 weeks [101,102]. Plasma exchange is recommended for therapy-refractory patients with fulminant disease [103,104]. Beyond treatment of the initial event, it is important to have a plan for long term follow-up to exclude a multiphasic disorder, which would warrant further diagnostic evaluation and a different therapeutic approach. 

## 4. Neuromyelitis Optica Spectrum-Disorder

Neuromyelitis optica (NMO) is an inflammatory disorder, traditionally considered monophasic, although relapsing cases have been described in which patients present optic neuritis and transverse myelitis [105]. NMO had been considered a variant of MS until an autoantibody against the water channel protein aquaporin-4 (AQP4), expressed abundantly on astrocyte end-feet, called AQP4-IgG (also called NMO-IgG), was discovered in patients with NMO, and found to be absent in patients with MS [106,107]. Incorporation of AQP4-IgG serology to revised NMO diagnostic criteria broadened the clinical and radiological spectrum of NMO [108]. The term NMO spectrum disorders (NMOSD) was introduced to include AQP4-IgG seropositive patients with limited forms of NMO, and at risk of future attacks, as well as patients with cerebral, diencephalic, and brainstem lesions, or coexisting autoimmune disease (e.g., systemic lupus erythematosus [SLE] or Sjögren syndrome [SS]) [109]. Accordingly, NMOSD was recognized as a humoral disease entity distinct from MS, and diagnostic criteria were revised in 2015 unifying the terms NMO and NMOSD [110].

Evidence supporting a pathogenic role of AQP4-IgG comes from different sources. Complement- as well as ab-dependent cytotoxicity [101,102] has been associated to AQP4-IgG, which when administered along with complement and/or pathogenic T cells, promotes development of NMOSD-like CNS lesions in rodents [111,112]. Inflammatory damage is characterized by astrocyte loss and deposition of both immunoglobulins and complement, followed by neutrophil, monocyte, phagocyte and eosinophil infiltration [113]. Importantly, AQP4 distribution coincides with deposition patterns of IgG, IgM, and products of complement activation present in active NMO tissue [114,115], and MRI lesions of patients with NMO overlap with sites of high AQP4 expression [116]. AQP4-IgG is believed to determine internalization of the glutamate transporter EAAT2, limiting glutamate uptake from the extracellular space into astrocytes, also resulting in oligodendrocyte damage and myelin loss [117]. Although most strongly expressed in the CNS, AQP4 is also present in the collecting duct of the kidney, parietal cells of the stomach, as well as in airways, salivary glands, and skeletal muscle [118]. However, peripheral organ damage does not typically occur, probably due to the presence of complement inhibitory proteins in these secondary target organs [119].

Despite caveats in knowledge on NMOSD epidemiology, prevalence has been estimated depending on the study population at 0.1–4.4 cases/100,000 individuals, and annual incidence at 0.20–4.0 per 1,000,000 [120,121]. Initial clinical manifestations occur at around 40 years of age, although children and the elderly account for 18% of cases. Female/male predominance is around 9:1, but not in children, where equal gender distribution has been observed [32,122]. 

According to the most recent diagnostic criteria, core clinical characteristics can involve 1 of 6 CNS regions, namely: optic nerve, spinal cord, area postrema of the dorsal medulla, brainstem, diencephalon, or cerebrum [110]. Clinical presentation particularly suggestive of NMOSD diagnosis includes: bilateral ON involving the optic chiasm with poor recovery compared to MS-ON, complete spinal cord syndrome determining paroxysmal spasms, and area postrema clinical syndrome characterized by intractable hiccups, or nausea and vomiting. No single clinical characteristic is pathognomonic of NMOSD, however [110]. In AQP4-IgG seronegative patients, diagnostic criteria are more rigorous. Patients must present at least 2 of the core clinical characteristics, and at least one of these must be: ON, longitudinally extensive transverse myelitis (LETM), or area postrema syndrome.

Given the focus of this review, in the following sections, only aspects related to NMOSD-related to spinal cord involvement will be addressed. 

Acute transverse myelitis symptoms in NMOSD patients (motor, sensitive, and frequently sphincter) are usually severe and bilateral, and recovery is incomplete compared to MS. Although overlap of clinical characteristics in MS and NMOSD myelitis does occur, symptom magnitude and disease history frequently contribute to establish differential diagnosis [30,32,120], as do certain MRI findings. LETM is the most specific neuroimaging characteristic found in NMOSD, and is uncommon in MS (Figure 2) [108]. Mirroring severe underlying tissue damage, lesions are generally hyperintense on T2-weighted, and hypointense on T1-weighted sequences [30]. Extending over three or more complete vertebral segments, they tend to localize in the center of the cord, because of the abundant AQP4 channel expression in grey matter. Lesions will usually occupy over 50% of the cross-sectional surface area of the spine, representing a complete, rather than incomplete, form of transverse myelitis which is more characteristic in MS. However, they also may be lateral, anterior, or posterior over the length of the lesion and be accompanied by cord swelling. The latter, when present, can generate concern over presence of a spinal cord tumor [123]. Chronic necrosis caused by NMOSD can in some cases result in spinal cord cavitation and cystic myelomalacia. Small areas of strong hyperintensity, higher than that of the surrounding cerebrospinal fluid (CSF), so-called bright spotty lesions, may be observed and could be useful to distinguish NMOSD from MS [124]. Acute NMO lesions extensively enhance following IV gadolinium administration. Lens-shaped ring-enhancement is detected in up to 32% of NMOSD patients [29,125,126]. Rostral extension of cervical lesions to the area postrema is another characteristic of NMOSD and can be helpful to distinguish it from other causes of longitudinal extensive myelopathy such as sarcoidosis, spondylotic myelopathy with enhancement, dural arteriovenous fistula, spinal cord infarct, and paraneoplastic myelopathy [127]. Although LETM is the most frequent form, 7–14% of NMO-myelitis involve <3 vertebral segments. However, short forms of NMO-myelitis are followed by LETM in ninety percent of cases. Short cord lesions should be suspected in patients with tonic spasm, coexistence of autoimmune disease, grey matter involvement and absence of OCB. As in MS, in 7–14% of cases, variation in presentation will be linked to time at which MRI scans are obtained [128,129,130]. Lesions limited to less than three segments will be detected at the beginning of disease or during remission [131]. In contrast, patients with longstanding disease may present short but coalescing lesions suggesting a LETM pattern [22]. Presence of a longitudinally extensive segment of cord atrophy is another characteristic finding in support of prior NMOSD myelitis [131]. 

Although in NMOSD the relationship between spinal cord atrophy, disease activity and disability is not fully known, two observations deserve mention. First, NMOSD patients predominately show spinal cord atrophy with only mild brain atrophy, while MS patients demonstrate more brain atrophy, especially in gray matter, suggesting a different underlying pathogenic mechanism [132]. Second, spinal cord atrophy can occur in patients without a clinical history of myelitis or visible spinal cord lesions on MRI, suggesting cord atrophy may be due to a diffuse underlying process. Alternatively, or perhaps in co-contributory fashion, patients may have experienced transient or subclinical inflammatory events not evident on conventional MRI [133].

Serum AQP4-IgG assay is the most useful test for NMOSD diagnosis. Based on criteria proposed by the International Panel for NMOSD, approximately 73–90% of patients with NMOSD express AQP4-IgG [134,135]. A cell-based assay (CBA) is recommended whenever possible because of its higher sensitivity (76.7%) and very low false-positive rate (0.1%) [136,137]. Indirect immunofluorescence assays and ELISA have less sensitivity (63–64% each), and can yield false-positive results (0.5–1.3% for ELISA) particularly at low titers [135,137]. Ultimately, 10–27% of patients with typical clinical and radiological features of NMOSD will not have detectable AQP4-IgG despite use of the best available assay. Lack of a diagnostic biomarker makes management of these patients more challenging especially of patients with monophasic disease [121,136]. Notably, using CBA, approximately 15–40% of AQP4-IgG seronegative NMOSD patients have been reported to have detectable antibodies against myelin oligodendrocyte glycoprotein (MOG) [137,138]. Aside from causing optico-spinal disease resembling NMOSD, anti-MOG antibodies have been identified in patients with clinical characteristics unlike those of patients with AQP4-IgG [32,137,139] (see below), suggesting a different underlying pathogenesis. Occasionally, patients without detectable serum AQP4-IgG are later found to be positive, possibly related to assay timing (antibody levels increase during exacerbations), or to impact of immunosuppressive treatment. 

Serum AQP4-IgG concentration is much higher than that found in CSF. The hypothesis behind this is that most AQP4-IgG is produced in peripheral lymphoid tissues and that a favorable serum/CSF antibody gradient is needed for penetration into the CNS, a concept supported by the fact that commercial CBA and flow cytometry detection of AQP4-IgG is more sensitive in serum than in CSF. Serum is therefore the optimal specimen for AQP4-IgG testing [140].

Some patients with NMOSD produce other autoantibodies in addition to AQP4-IgG, as occurs in patients with SLE or SS [118]. Since LETM has also been described in patients with these conditions, the possibility exists that NMOSD symptoms arise secondary to SLE or SS. Limited existing data in this regard shows that in such patients, AQP4-IgG detection rates are similar to those observed in patients with NMOSD without associated rheumatic disease, suggesting LETM in NMOSD is not secondary to SLE or SS, and these patients suffer from two independent, coexisting autoimmune diseases [118,141,142,143].

CSF pleocytosis (>50 cells/µL) or presence of neutrophils or eosinophils during NMOSD attacks may help to distinguish NMOSD from MS [123,137].CSF OCBs are usually absent, although they may sometimes be transiently detectable during an attack [123,144].Given the high morbidity associated with NMOSD exacerbations, the goals of pharmacotherapy are to aggressively treat acute attacks, (including the initial episode) and prevent future relapses, minimizing CNS damage and long-term disability [145,146]. Different pathophysiologic mechanisms are known to characterize MS and NMOSD, a finding at least partially demonstrated by the fact that exacerbations can be precipitated by fingolimod, IFNβ and natalizumab, treatments that are effective in MS. Aside from the need for accurate diagnosis, evaluation of occult infection or metabolic disturbances should be carried out to identify pseudo-relapses Although there are no randomized controlled trials in large cohorts examining treatment of acute relapses, NMOSD exacerbations are typically treated with 1 g of IVMP for 3–5 consecutive days [147,148]. Severe NMOSD relapses or patients who do not respond to treatment with IVMP may benefit from plasma exchange (PLEX) [72,147,148,149]; which targets specific antibodies, complement and several pro-inflammatory proteins [150] Early (≤5 days), aggressive treatment with PLEX is linked to better outcome [151]. Interestingly, positive results of PLEX are obtained both in seropositive as well as seronegative NMOSD patients [152,153,154]. In order to avoid relapses, different immunosuppressive strategies are used in daily neurological practice including: oral corticosteroids, mycophenolate mofetil or azathioprine (both oral purine analog anti-metabolites), rituximab (IV anti-CD20 monoclonal antibody) and tocilizumab (anti-IL-6 receptor monoclonal antibody [146,147]. However, none of these agents have been specifically approved for NMOSD treatment, and off-label use has arisen based almost entirely on results from uncontrolled observational studies [146,147]. Recently, three new monoclonal antibodies with different mechanisms of action and routes of administration have shown efficacy in NMOSD patients: eculizumab (anti-complement protein C5) [155], inebilizumab (anti-CD19) [156], and satralizumab (anti-IL-6R) [157], significantly reducing risk of new relapses compared to placebo, particularly in AQP4-ab-positive patients, with clinical stabilization or improvement in most cases. All these drugs demonstrated good safety and tolerability profiles with limited side effects. Future evaluation in real-life studies will be needed though, to estimate annual relapse rates and compare results to those of older drugs. 

## 5. Myelin Oligodendrocyte Glycoprotein Antibody-Associated Disease

Myelin oligodendrocyte glycoprotein, a member of the immunoglobulin superfamily, is exclusively expressed on the surface of oligodendrocytes and on the outermost lamellae of myelin sheaths in the CNS. Given its structure and location it could potentially function as a cell surface receptor, or cell adhesion molecule. Furthermore, its extracellular location makes it a target for autoimmune ab- and cell-mediated responses, in inflammatory demyelinating diseases. Interesting results from animal studies on MOG ab-associated demyelination lead to this antibody being considered a marker for MS [158,159]. However, subsequent studies in large populations of MS patients found seropositivity prevalence in this condition was similar to that detected in other inflammatory neurological diseases, as well as to levels in control subjects, generating skepticism over whether these ab could be considered a true biomarker of MS [160,161,162]. Seminal studies using murine anti-MOG ab have highlighted the fact that ab target epitopes of native MOG are biologically relevant in their conformational state, rather than in linearized or denatured MOG. Therefore, CBA, which maintains the native conformational form of the extracellular portion of MOG, is the most recommended technique to study ab levels.

There is current international consensus that anti-MOG ab are important in both pediatric and adult demyelination. Different research groups have identified seropositive MOG ab populations in children with ADEM, particularly in recurrent forms of the disease [163,164,165,166]. Other studies later confirmed presence of MOG ab in 25% to 30% of AQP4 seronegative NMOSD patients with recurrent ON. Substantial differences between both diseases in histopathology, as well as in vivo and vitro studies demonstrating a direct pathogenic role for MOG-IgG, suggest it represents a separate individual entity. Anti-MOG ab are already present at disease onset, both in serum and CSF, in some patients, persisting also during remission in the majority of patients, which argues against anti-MOG ab presence as a secondary epiphenomenon [167,168,169,170]. Notably, serum anti-MOG ab detection is more sensitive than CSF assay. 

Since these observations, an increasing number of patients with diverse phenotypes related to these antibodies have been described. A comparison of patients with MOG ab disease to AQP4 NMOSD cases showed the former were younger [68,169,170,171], did not show significant female predominance [172], and were more commonly Caucasians; whereas AQP4-seropositive NMOSD was found predominantly in non-Caucasian populations [173,174].

The most commonly reported presentation of anti-MOG ab-associated disease is ON, which can be bilateral and recurrent in up to 61% of cases. Interesting, imaging of the optic nerve frequently shows peri-optic nerve sheath contrast enhancement, extending into the surrounding soft tissue, a radiological characteristic not observed in MS or AQP4 positive patients [175,176]. 

Approximately half the patients with MOG ab-associated disease present episodes involving the spinal cord [177,178]. The most common symptoms include paraparesis, and sensory and sphincter dysfunction. On MRI, LETM is frequent and short myelitis less common. Any segment of the spinal cord can be affected, although lesions are more frequent in the thoracolumbar and/or conus medullaris regions, as opposed to the more common cervicothoracic involvement observed in AQP4 ab positive and MS myelitis cases [178,179]. Anti-MOG ab associated myelitis is hyperintense on T2-weighted and iso-hypointense on T1-weighted sequences, showing contrast enhancement during acute phases in up to 70% of cases [172]; Figure 3 and Figure 4. MOG ab-related disease does not commonly result in cord necrosis or cavitations as observed in AQP4-mediated cases [134,175,178]. Due to the predilection for conus localization, bladder, bowel, and erectile dysfunction is observed in approximately 70% of patients [167]. In comparison to AQP4-IgG^+^ NMOSD, MOG ab disease myelitis appears to more focal and with better clinical outcome, although poor outcome with permanent disability has been described for both conditions [156]. Notably, anti-MOG ab serum titers follow disease activity levels, with significantly higher concentration during acute attacks than remission, further supporting the concept of their pathogenic role [172]. 

Although ON and myelitis are the two most frequent forms of presentation of anti-MOG ab disease, coexistence of brain, brainstem, or cerebellar involvement is frequent, and may even be extensive. Nausea, vomiting, and respiratory disturbances are some of the symptoms that can be present in cases of brainstem involvement [177]. 

Different study groups have developed MRI diagnostic criteria to differentiate MS, from NMOSD and from anti-MOG ab-associated disease, showing 91% sensitivity distinguishing MS from AQP4+ NMOSD, and 95% from anti-MOG ab-associated disease [173,179]. More recently, the criteria were subtly modified to include spinal cord in the analysis, increasing sensitivity to 100% and specificity to 79%, reflecting the crucial importance of spinal cord findings in anti-MOG-ab disease. Interestingly, this radiological criterion was particularly useful in patients with ON, a clinical presentation common to all three diseases [180].

Patients with anti-MOG ab-associated disease were initially described as experiencing a monophasic disease [91,140,178]. However, recent studies found a high proportion of patients presenting relapsing disease [173,181]. Anti-MOG ab-positive patients exhibited better motor and visual outcome compared to AQP4-IgG positive patients after the first episode [170,181]. 

Anti-MOG ab are present in approximately 40% of children with ADEM. In this group, most patients develop LETM, and similar to patients without anti-MOG ab, show large, ill-defined, bilateral lesions in the brain, which typically resolve completely, in correlation with improved clinical outcome [165,177].

MOG ab-positive patients show rapid response to steroids and plasma exchange [177], but tend to relapse quickly after steroid withdrawal or cessation [182,183]. Therefore, slow steroid taper is recommended to minimize chances of early relapses. In adult patients, persistent seropositivity following initial treatment and clinical resolution is one of the main reasons to consider long term immunosuppression with steroid-sparing agents including mycophenolate, azathioprine or rituximab [135,169,170,184,185,186]. The significance of this finding is less clear in pediatric patients with ADEM and persistence of serum anti-MOG abs.

## 6. Glial Fibrillary Acid Protein Antibody-Associated Myelitis

A novel autoimmune CNS disorder characterized by the presence of antibodies specific for glial fibrillary acidic protein (GFAP) has recently been described. In the largest series published to date, median symptom onset age was around 40 years, with similar incidence in both women and men [187,188]. All patients with GFAP-IgGs reacted against the mature (α) GFAP isoform, with only a few patients showing immunoreactivity against the immature (ε) isoform [188]. GFAP is a cytoplasmic protein not accessible to IgG in intact cells, therefore, it is possible that immune cells also contribute to the tissue damage observed in this condition, for example GFAP peptide-specific CD8^+^ T lymphocytes [189]. Eventually other immune cells sensitive to steroids, such as microglia and macrophages, can also play a role in the disease, acting directly, or through the release of molecules modulating the immune response such as cytokines or chemokines [187,190,191,192].

Clinical phenotype of GFAP-IgG astrocytopathy is heterogeneous and still poorly defined. The predominant clinical syndrome includes meningitis, encephalitis, and myelitis, or all three (meningoencephalomyelitis) with or without optic disc edema [188,193,194]. 

Myelitis occurs in up to 68% of patients with GFAP-IgG. However, its presentation as isolated clinical manifestation is infrequent. Despite the fact that autoimmune GFAP astrocytopathy and NMOSD-related myelitis share some clinical features, certain differences are worth mentioning [195]. Influenza-like prodromal symptoms and bowel/bladder dysfunction are common features in GFAP-IgG myelitis, while numbness and weakness followed by tonic spasms, frequent NMOSD symptoms, are rare. Notably, sensory level and Lhermitte’s phenomenon are usually absent in GFAP-IgG myelitis, which is found in the cervical or thoracic spinal cord, in central location, usually involving at least three vertebral segments [195]. Lesions are hyperintense on T2-weighted sequences and may show a thin and linear pattern of contrast enhancement along the course of the central canal, different to the patchy or ring-like contrast uptake seen in NMOSD [187]. GFAP-IgG lesions have poorly-defined margins and less cord swelling compared to AQP4-IgG myelitis [195]. Short myelitis has also been reported in association with brain symptoms [187,194,195].

Notably, brain MRI findings significantly contribute to discriminate GFAP-IgG from other pathologies. A striking pattern of linear radial periventricular contrast enhancement is highly specific for GFAP-IgG-associated disease. Similar radial enhancement patterns have been described in the cerebellum in a lower percentage of patients [184,185,193].

Anti-GFAP abs can be detected in serum in 45% of patients, but sensitivity increases to 92% when ab are assayed in CSF [187]. Up to 50% of cases coexist with *N*-methyl-d-aspartate receptor (NMDAr) antibodies or anti-AQP4 ab, and up to 34% of patients may present concomitant neoplasms, with ovarian teratoma as the most prevalent [187]. These associations explain the diverse phenotypes reported [187,188,194]. Marked elevation of white cells and protein are common findings in CSF, and intrathecal oligoclonal bands may be present in 50% of patients [187].

Most reported GFAP-IgG cases show improvement in clinical, radiological, and CSF abnormalities after receiving high-dose intravenous methylprednisolone for 3–5 days [184,192]. Although nonresponsive-patients have been described, need for plasma exchange is significantly less frequent compared to patients with NMOSD [193,195,196]. In one study, 50% of patients with long-term follow-up (>24 months) had a relapsing course, 27% had a monophasic course and 23% had progressive disease in spite of adequate treatment. Clinical relapses were frequently associated with recurrent gadolinium enhancement on MRI and elevated CSF white cell count, with further remission observed after restarting steroids [187].

GFAP-IgG is unlikely to be directly pathogenic, as GFAP is an intracellular protein. However, it could be an excellent biomarker, identifying a neoplasm early on, leading to prompt and efficient treatment and prevention of long-term disability in GFAP-IgG myelitis cases.

## 7. Conclusions

Overall, demyelinating myelopathies belong to a complex and heterogeneous group of diseases, in which differential diagnosis can be difficult (Table 1). Clinical features, time-course, CSF characteristics, specific serum assays, and brain and spinal cord MRI findings all contribute to determine diagnosis, select the best treatment option and establish prognosis for each subtype. Early treatment with IV steroids and PLEX is accepted in all etiologies, but more specific treatment strategies may subsequently be adopted based on final diagnosis.

## Figures and Tables

**Figure 1 biomedicines-08-00130-f001:**
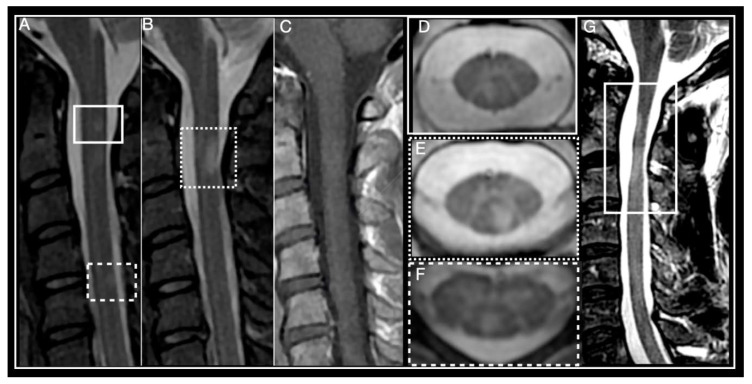
Multiple Sclerosis myelitis. (**A**–**F**) 32-year-old woman diagnosed with relapsing remitting course (RRMS) 2 years earlier, EDSS 0. (**A**,**B**) Sagittal short-tau inversion recovery (STIR) showing small, focal, chronic, peripheral lesions. (**C**) Sagittal post-contrast T1 weighted, absence of enhancement, T2 lesions are isointense. (**D**–**F**) axial T2 multiple-echo recombined gradient echo (MERGE). (**D**) right paramedian posterior lesion corresponds to lesion framed by a box in (**A**). (**E**) left paramedian posterior lesion corresponds to lesion framed by a dotted box in (**A**). (**F**) posterior lesion corresponds to lesion framed by a dotted line in (**A**). (**G**) 46-year-old man diagnosed with primary progressive multiple sclerosis (PPMS) in 2011, EDSS 6. Sagittal T2-weighted, framed area shows multiple sclerosis (MS) lesions and spinal cord atrophy.

**Figure 2 biomedicines-08-00130-f002:**
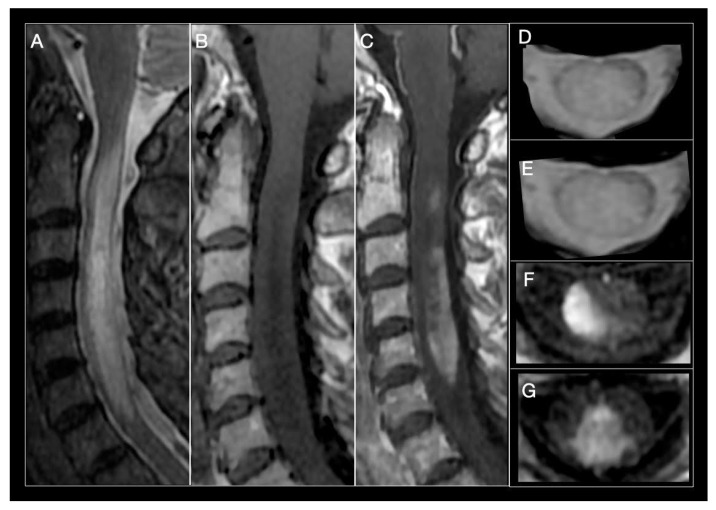
Neuromyelitis optica (NMO) myelitis. Images from a 58-year-old woman with acute longitudinally extensive myelitis (C1–C7). (**A**) Sagittal STIR showing an extensive lesion, involving more than 3 segments, that widens the cervical spinal cord. (**B**) Sagittal T1-weighted sequences show an extensive T1-hypointense lesion. (**C**) T1-weighted images after contrast administration, extensive enhancement of cervical lesion. (**D**,**E**) Axial T2-MERGE hyperintense area that involves more than half the diameter of the spinal cord. (**E**,**F**) Axial T1-weighted, intense contrast enhancement of lateral (**E**) and central-posterior (**F**,**G**) areas.

**Figure 3 biomedicines-08-00130-f003:**
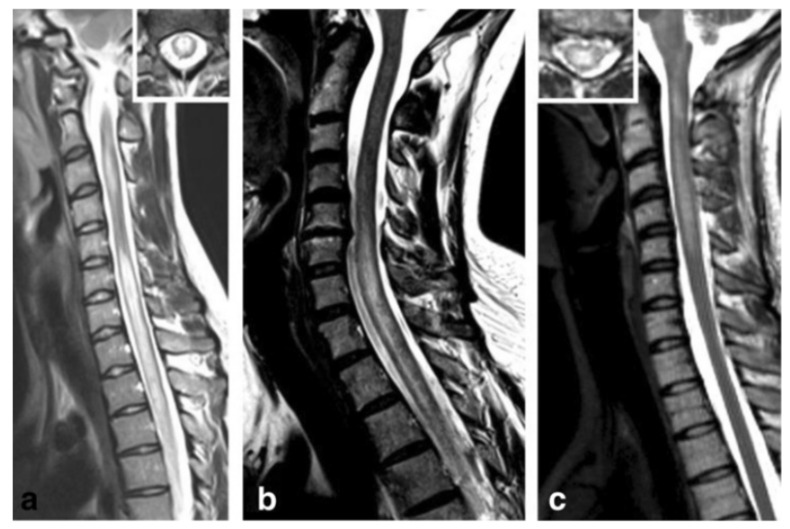
Anti-myelin oligodendrocyte glycoprotein (MOG) antibody myelitis. (**a**) Sagittal T2-weighted spinal MRI performed at disease onset revealed a large longitudinal centrally-located lesion extending over the entire spinal cord, as well as swelling of the cord. (**b**) Longitudinally extensive central spinal cord T2 lesion in another patient. (**c**) T2-hyperintense lesions extending from the pontomedullary junction throughout the cervical cord to C5, in a third patient. Insets in (**a**) and C show axial sections of the thoracic cord at lesion level [172]. Figure is extracted from Jarius, S. et al., *J Neuroinflammation* 2016, *13*, 280 (http://creativecommons.org/licenses/by/4.0/).

**Figure 4 biomedicines-08-00130-f004:**
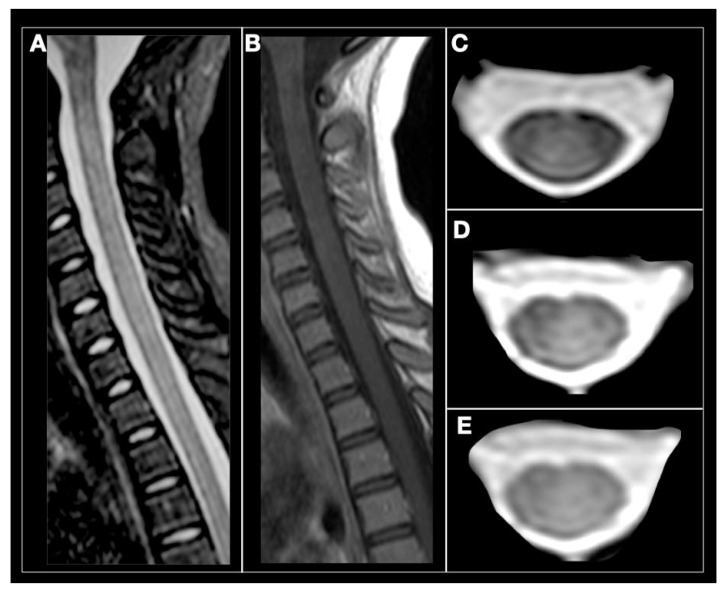
Anti-MOG antibody myelitis. A 12-year-old girl with relapse in the cervical spine. (**A**) sagittal STIR, subtle and diffuse hyperintensity of the cervical spinal cord. (**B**) Sagittal T1-weighted, spinal cord is isointense without contrast enhancement. (**C**–**E**) axial T2-weighted images showing subtle and diffuse spinal cord hyperintensity (Courtesy Dr. Angeles Schteinschnaider).

**Table 1 biomedicines-08-00130-t001:** Main features in demyelinating myelopathies of different etiology.

	MS	ADEM	NMOSD	MOG-IgG Disease	GFAP-IgG Disease
Estimated F:M ratio	3:1	1:1	9:1	1.3:1	1:1
Age * (yrs)	30	6	37	33	40
Myelitis clinical features	Sensory loss, gait impairment, weakness, sphincter involvement	Transverse myelitis	Transverse myelitis	Paraparesis, sensory symptoms and sphincter involvement	Sensory symptoms, sphincter disfunction
Clinical course	Relapsing (85%) or progressive (15%)	Typically monophasic (69–90%)	Relapsing (90%)	Monophasic (58%) or relapsing (42%)	Relapsing (50%), monophasic (27%) or progressive (23%)
Serology findings	Not relevant	Not relevant	Serum AQP4-IgG. coexistence with other systemic disease antibodies (ANA, SSA or SSB).	Serum MOG-IgG	Anti-GFAP ab + in serum or CSF (Serum Anti-AQP4-IgG and/or anti-NMDAr ab coexistence,
Presence of OCB	80–95%	0% to 29% (usually transient)	Up to 30% (usually transient)	Up to 12%	Up to 50%
CSF	Generally normal or mild inflammatory changes	Mild pleocytosis and increased proteins up to 62%	Pleocytosis (neutrophils and eosinophils can be found) and mild elevated proteins	Normal or slightly inflammatory changes	Marked elevation of white blood cells and elevated protein levels
Brain MRI	Dawson fingers, lesions perpendicular to ventricles Cortical/yuxtacortical lesions Perivenular Nodular or ring/open-ring enhancing lesions Unilateral short optic nerve enhancement	Subcortical or deep gray matter bilateral, sometimes poorly-defined Simultaneous enhancement with gadolinium	Periependimal lesions Tumefactive lesions Involvement of corticospinal tract Marked enhancement, ‘cloud like’ Bilateral, long optic nerve enhancement	Non—specific supratentorial subcortical or small deep white matter foci. Occasionally T2 lesions in brainstem, and infratentorial regions Anterior bilateral ON with perineural optic nerved enhancement	Linear radial periventricular contrast enhancement pattern
Spinal cord MRI	Small, peripheral, posterolateral lesions Less than 3 segments Gadolinium enhancement during acute phase	LETM or multiple short segment myelitis Edematous lesions and gadolinium enhancement in acute phase	Central LETM Edematous Necrosis or cavitation Gadolinium enhancement in acute phase	LETM or short myelitis, frequent conus medullaris involvement Linear gadolinium enhancement of the ependymal canal	LETM Central lesions

Ab: antibodies, ADEM: acute disseminated encephalomyelitis, AQP4: Aquaporin 4, F: female, GFAP: glial fibrillary acid protein, LETM: longitudinally extensive transverse myelitis, M: male, MOG: myelin oligodendrocyte glycoprotein, MRI: magnetic resonance imaging, MS: multiple sclerosis, NMDAr: *N*-Methyl-d-aspartate receptor, NMOSD: neuromyelitis optica spectrum disorder, OCB: oligoclonal bands. * estimated media.

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
