# Peer review of "Spinal Cord Involvement in MS and Other Demyelinating Diseases"

_biomedicines, 2020, doi:10.3390/biomedicines8050130_

Round 1

Reviewer 1 Report

          The authors summarize the current research on several demyelinating myelopathies, with a focus on imaging diagnostics.  The review does an excellent job of introducing, comparing, and contrasting MS, ADEM, NMOSD, MOG-IgG disease, and GFAP-TgG disease.  There are some minor writing errors, but the manuscript is otherwise solid.  My only concern is that with the in-depth focus on imaging, this review cannot critically review the majority of the science.  A reviewer expert in imaging and radiology should also provide support by reviewing.

Major Concerns

  1. The manuscript would benefit greatly from critical review by an expert in imaging and radiology since this is the focus of the majority of the summarized research.

Minor Concerns

  1. Despite the focus on imaging in the spinal cord, the spinal cord is not mentioned in the abstract.
  2. Line 13 page 1 has a pronoun with unclear subject. This should instead state “these subtypes.”
  3. Lines 14-18 page 1 have an extraordinarily long sentence that should be split for clarity.
  4. Line 15 page 2 should state “susceptibility” genes not “susceptible.”
  5. Several areas of the manuscript have inconsistent fonts or font sizes including line 38 p1, line 36 p2, and line 28 p7.
  6. Line 17 page 5 states an observation of absence, which is improper pairing of these terms. Perhaps instead state “no triggering event can be observed.”

Author Response

Replies to the Reviewer 1

  1. The manuscript would benefit greatly from critical review by an expert in imaging and radiology since this is the focus of the majority of the summarized research.

Reply: One of the authors of the manuscript is Dr. María Inés Gaitán, who is an expert in neuroimaging in general and in demyelinating diseases in particular, with extensive training at

NIH.  As co-author, Dr. Gaitán wrote certain sections and reviewed and edited the original manuscript. Nevertheless, as requested by the reviewer, the manuscript has been revised and additional paragraphs specifically referring to radiological findings were incorporated, particularly to the Multiple Sclerosis section. These had been previously omitted, to reduce manuscript length, and maintain greater balance vis a vis the other diseases discussed in the manuscript. We hope the new additions to the text meet the Reviewer's expectations. Please see page 6, lines 15-17; page 7 lines 23-25; page 8 lines 3-4: page 8 lines 10-13 and 22-24; page 9 lines 1-2, 13-20 and 23-25; page 10, 1-10.

  1. Despite the focus on imaging in the spinal cord, the spinal cord is not mentioned in the abstract.

Reply: The reviewer is correct in this observation. The word "spinal cord" has been incorporated to the abstract. Please see abstract page 2 line 7.

  1. Line 13 page 1 has a pronoun with unclear subject. This should instead state “these subtypes.”

Reply: We appreciate the Reviewer's observation. The sentence has been modified in line with his/her suggestion.

  1. Lines 14-18 page 1 have an extraordinarily long sentence that should be split for clarity.

Reply: As suggested by the Reviewer the sentence has been rewritten. Please see abstract lines 6-12.

  1. Line 15 page 2 should state “susceptibility” genes not “susceptible.”

Reply: We appreciate the Reviewer's observation. The sentence has been corrected according to his/her suggestion

  1. Several areas of the manuscript have inconsistent fonts or font sizes including line 38 p1, line 36 p2, and line 28 p7.

Reply: We appreciate the Reviewer's observation. We have edited font inconsistencies referred to by the reviewer.

  1. Line 17 page 5 states an observation of absence, which is improper pairing of these terms. Perhaps instead state “no triggering event can be observed.”

Reply: Once again we agree with the Reviewer's observation and have corrected the sentence accordingly.

Reviewer 2 Report

In this Review Manuscript, the Authors describe characteristics, pathophysiology, clinical and MRI findings and treatment options of the spinal cord involvement in Multiple Sclerosis (MS) and other demyelinating disorders including Neuromyelitis Optica Spectrum Disorders (NMOSD), Acute Disseminated Encephalomyelitis (ADEM), anti- Myelin Oligodendrocyte Glycoprotein (MOG)-antibodies (ab) associated disease, and Glial Fibrillary Acidic Protein (GFAP)-IgG associated disease, to provide guidance in the diagnosis of these conditions. Indeed, diagnostic accuracy in myelopathies is poor and represents a challenge for neurologists in daily management of these patients.

The Manuscript is well structured and clear:

  • An adequate amount of literature has been examined
  • Each disease is well described in terms of epidemiologic characteristics, pathophysiology, clinical and MRI findings, treatment options and prognostic implications.
  • The focus on spinal cord aspect is clear for each disease
  • The main features in demyelinating myelopathies of different etiology are well summarized in Table 1

As a general comment I think this Review clearly describes the spinal cord involvement in different myelopathies (MS, NMOSD, ADEM, anti-MOG associated disease anti-GFAP associated disease), and it also summarizes clinical, MRI and diagnostic characteristics that can help neurologist in daily management of these patients.

I have the following suggestions:

  • The Authors should provide the criteria they used to perform the literature research for this review (key terms, number of papers etc). I think this always helps the reader to better understand the Authors’ work.
  • Page 12, Raw 47: please provide a Title for this Chapter. This is a sort of Conclusion, it is not part of Chapter 6.  

Author Response

Replies to the Reviewer 2.

  1. The Authors should provide the criteria they used to perform the literature research for this review (key terms, number of papers etc). I think this always helps the reader to better understand the Authors’ work.

Reply: We appreciate the Reviewer's comment. The criteria used for the literature search has now been included in the revised manuscript. We describe sources, publication time period of articles and key terms applied for the search. Please see page 4 lines 7-12.

  1. Page 12, Raw 47: please provide a Title for this Chapter. This is a sort of conclusion, it is not part of Chapter 6.

Reply: We appreciate the Reviewer's observation. The chapter has been corrected according to his/her suggestion. Please see page 34 line 1.